# Willingness and Influencing Factors of Farmers' Forestland Management in Ethnic Minority Areas: Evidence from Southwest China

Ya Li [1,†], Haiqing Chang [1,†], Yaquan Dou [2,†] and Xiaodi Zhao [2,3,*]

1 College of Economics and Management, Southwest Forestry University, Kunming 650224, China; liy@swfu.edu.cn (Y.L.); changhq@swfu.edu.cn (H.C.)
2 Research Institute of Forestry Policy and Information, Chinese Academy of Forestry, Beijing 100091, China; douyq@caf.ac.cn
3 Faculty of Forestry, The University of British Columbia, Vancouver, BC V6T 1Z4, Canada
* Correspondence: zhaoxiaodi@caf.ac.cn; Tel.: +86-136-7105-3260
† These authors contributed equally to this work.

**Abstract:** This paper uses a questionnaire and interviews from households in ethnic minority areas of the Jianchuan County (Dali Bai Autonomous Prefecture) and Pingbian County (Honghe Hani and Yi Autonomous Prefecture) in Yunnan Province to explore the willingness of foresters to manage forests. Using the Sustainable Livelihoods Analysis framework, we select three indicators including the variables of individual social economic attributes, the cognition and experience of forest landowners, and policy guidance. We use a binary logistic regression model to analyze the factors affecting the willingness of foresters to participate in forest management. Through the above analysis, we found the following: (1) Forest landowners' willingness to engage in forest management in ethnic minority regions is relatively high, at 71.98%. (2) Variables of individual social economic attributes have the most significant degree of influence on the willingness to engage in forest management. (3) Standard of living and the woodland area have a significant positive effect on forest land management intentions, while education level, whether they are compensated by public welfare forests, and whether they have participated in the project of returning farmland to forest and grassland have a significant negative effect on management intentions. (4) There are significant differences between forest landowners' willingness to engage in forest management and the influencing factors between minority regions and non-minority regions.

**Keywords:** forest landowner; management willingness; forest ecological products; ethnic minority areas; Sustainable Livelihoods Analysis framework

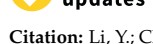



## 1. Introduction

The relationship between economic development and ecological protection has always been a key issue of global concern [1]. With the increasing severity of global climate change and biodiversity loss, researchers are currently exploring which is the best solution to alleviate the contradiction between economic development and ecological conservation [1–3]. Forest resources, as the main body of terrestrial ecosystems, are the most important ecological foundation for the survival and development of human society. Additionally, they play an irreplaceable role in maintaining the global ecological balance, guaranteeing ecological security and improving human living environments [4]. In recent years, the conservation and development of global forest resources have received increasingly widespread attention from international organizations, national governments and the public. In response to the impact and challenges of a series of global problems, attaching importance to forests and protecting ecology have achieved the broad consensus of the international community and become the national strategies of various countries. As an important carrier of the

protection of forest resources and the development of the forestry industry, the use of forest land plays an important role in alleviating the contradiction between ecological protection and economic development. On the one hand, forest land, as an important part of forest resources, is an important basis for ecological environment quality and carrying capacity. It is also an important foundation for forest existence and ecological restoration [5]. On the other hand, forest land, as an important production factor for the development of the forestry industry, represents important natural capital needed to promote farmers' income [6]. Simultaneously, as the main players in forest land management and utilization, forest landowners play an important role in the protection and utilization of forest resources, and their management intentions and behaviors will affect the protection of forest resources and the development of the forestry industry. Therefore, foreign scholars have gradually started to focus on the willingness of forest landowners to engage in [7–10], and their behavior [11–14] towards, forest management. Price (1997) concluded that factors such as forest land resource endowment and individual farmers' characteristics have a certain degree of influence on the transformation of management behavior and willingness through the efficiency of forestry production in the UK [15]. Viitale (1998) found that reducing the input to public benefit forests can appropriately change the generally low technical efficiency of production [16]. Denis J (2011) suggested the importance of policies for developing forestry [17], and Thant (2011) studied the role and influence of the willingness and behavior of 200 households in Myanmar to achieve sustainable forestry [18]. Through a study of selected African American forest landowners in the southern US, Goyke (2019) found that professional advice had the greatest degree of influence on forest landowners' participation in forest management behavior [19]. Jang-Hwan Jo et al. (2019) conducted a statistical analysis based on panel data from sustainable forest land management institutions in Korea and found that a number of elements related to the livelihood strategy level influence farm household forestry income to varying degrees, and thus also affects the willingness to engage in forest management and the sustainability of forestry [20].

Collective forests are China's important ecological barrier and the supply base of forest products. They can ensure national timber and food security, cope with climate change, and consolidate and improve the results of poverty alleviation. In order to take full advantage of the role of forest resources in ecological security and food security, and to effectively stimulate the enthusiasm of forest landowners to engage in forest management, the General Office of the CPC Central Committee and the State Council in China issued the "Opinions on Comprehensively Promoting the Reform of Collective Forest Rights System". The "Opinions" determined the foresters' rights to use and engage in forest management and their ownership of forest trees, and foresters gained the autonomy to engage in forest management. Through the collective forest reform, the cultivation of collective forest resources has been strengthened, and the forest stock of collective forests nationwide has increased by nearly 2.4 billion cubic meters compared with that before the forest reform. The transfer of collective forest rights has been steadily promoted, and the number of new business entities reached 294,300, operating more than 18.667 million hectares of forest land [21]. In recent years, although the reform of the collective forest rights system has achieved good results, the productivity of collective forest land has not yet been fully achieved, the comprehensive benefits and operational efficiency of collective forests [22] are still not high, and the economic income from forestry for forest landowners is relatively small [23]. The enthusiasm of forest landowners and social capital to engage in forest management is not high, so how to pass the "last kilometer" to realize ecological beauty and the wealth of the people has become an urgent problem. At the same time, with the continuous promotion of the reform of the collective forest rights system, the willingness of forest landowners to engage in forest management has become the focus of academic research in the process of understanding the role of forest landowners. With the development of modern forestry, forest landowners' management of forest land has developed from decentralization and diversification to centralization and unification [24,25], and joint-family and moderate-scale operations can counteract the shortcomings of single-

family and decentralized operations in terms of technology, efficiency, and costs [26–30]. In addition, domestic scholars have found that there is often a gap between the willingness and behavior of forest landowners to engage in forest management. Forest landowners who show willingness to engage in forest management may not actually display an operating behavior, and there are many influencing factors for this conversion [26], such as forest land resource endowment [31], individual forest landowners' characteristics [32], policy compensation [33], and operating philosophy [34]. For example, Xie concluded that factors such as forest land resource endowment and individual farmers' characteristics, have some influence on the transformation of forest land management behavior and willingness through a study of 10 forest counties in Jiangxi Province [35].

Yunnan Province is not only an ecological barrier in the southwest region, but also a community of fate and responsibility with the ecology of the people of South and Southeast Asia. It strategically safeguards China's—and even international—ecological security [36]. Over the years, Yunnan Province has continuously strengthened cross-border biodiversity conservation and cultural exchanges with neighboring countries, such as Laos, Vietnam, and Myanmar, and held the "China–Myanmar Forest Resources Protection and Community Development Forum" and the "China–Myanmar Forestry Cooperation Group First Consultation". The Greater Mekong Subregion is a bridge that connects China's southwestern region and Southeast Asian countries. The effective utilization and protection of forest resources and ecological restoration have increasingly become a hot topic in the Lancang-Mekong River Basin, especially the poverty problem [37]. In addition, Yunnan, as a frontier province of the Lancang-Mekong poverty reduction cooperation, is actively engaged in poverty reduction strategies with Mekong countries and continuously promotes the sustainable development of forestry in the Greater Mekong Subregion. For example, Myanmar and Yunnan use bamboo and rattan as a forestry poverty reduction product [38]; northeastern Thailand [39] and northern Laos select ecotourism as a forestry poverty reduction breakthrough; Vietnam adopted a community forestry plan [40]; and Cambodia's community-based management of forestry [41] employs forestry resources to significantly reduce poverty. Yunnan in China and Thailand are the most effective countries or regions in poverty reduction in the Lancang-Mekong Basin through government-led or community-led approaches. However, due to historical reasons and production conditions, the fragmentation of land, the difficulty to engage in forest management, and the increase in costs, especially in Yunnan's ethnic minority regions, the low utilization rate of forest resources is considerable, which has hindered the sustainable development of forestry and the sustainable livelihood of forest landowners. From the perspective of resource economics, the "tragedy of the commons"—caused by the idle and excessive use of forest resources—results in similar inefficiency [42]. Therefore, in this context, the ways in which forest resources can be fully used and the willingness of forest landowners to engage in forest management have become the focus of research to promote the coordinated development of forestry ecology, economy, and society in minority regions. As the direct subject of the protection and utilization of forest land resources, the willingness of forest landowners affects the utilization and management efficiency of forest land and also has a certain impact on forestry industry development and ecological construction. Since the full-scale reform work in Yunnan Province in 2010, the general enthusiasm of farmers for forestation and forest protection has been generally high, promoting the rapid development of forestry industry with the development of special economic forests. It has changed the phenomenon that "foresters in collective forest areas are generally unwilling to reforest, the collective is unable to reforest, and the forestry department has no money to reforest" [43]. To date, the reform has given farmers the right to forest ownership and use in Yunnan Province, and the state has decentralized the right to operate and dispose of forest resources. At the early stage of reform, development with the goal of ecological priority lacked realism, and farmers blindly developed and exploited commercial forests in the pursuit of short-term interests, resulting in a large loss of forest land and forest resources within a short period of time. In the late reform period, government departments

promulgated a series of regulations on forest harvesting limits, policies on natural forest logging bans, and the implementation of corresponding natural forest protection projects, which made foresters more aware of ecological protection, but found that their enthusiasm and willingness to manage diminished. In this study, our aim is to give full play to the endowment of forest resources in ethnic minority areas of Yunnan Province, as well as to realize the industry-ecosystem virtuous cycle of forestry and the preservation and appreciation of natural forest assets. We explored the willingness of foresters to operate in Yunnan's ethnic minority areas in the context of reform, so as to stimulate the endogenous motivation of ethnic minority foresters to participate in the forestry industry and ecological construction. Using Sustainable Livelihoods Analysis (SLA) framework as the analytical framework, this study adopts some typical ethnic minority areas in Yunnan Province as the research object and answers the following questions by analyzing ethnic minority farmers' willingness to manage forest land in Jianchuan County (Dali Bai Autonomous Prefecture, Yunnan Province, China) and Pingbian County (Honghe Hani and Yi Autonomous Prefecture, Yunnan Province, China): Are there differences in the willingness of ethnic minority forest farmers and ordinary forest farmers to participate in forest management? How can the willingness of forest farmers in ethnic minority areas to participate in forest management be stimulated? What are the factors that influence farmers' willingness to manage forest land in ethnic minority areas? This paper also provides a reference for neighboring countries or regions of Yunnan Province regarding forest land management, ecological poverty reduction and industrial development.

## 2. Materials and Methods

### 2.1. Study Area Overview

Jianchuan County is a county seat largely considered the country's national garden, with a forest coverage rate of 74.5%, located in the northwest of Yunnan Province and the north of Dali Prefecture, with Heqing County (Dali Bai Autonomous Prefecture, Yunnan Province, China) in the east and Lijiang City (Yunnan Province, China) in the north. It has a mountainous area of nearly 90% of the territory, and 96.2% of the county's population is composed of minorities. The Bai people account for 91.2% of the total population (Figure 1). Jinhua and Shaxi are the two most populous townships (communities) in Jianchuan County, and are rich in woodland resources. The Jinhua area (Jinhua Town, Jianchuan County, Dali Bai Autonomous Prefecture, Yunnan Province, China) is dominated by public welfare forests, while Shaxi Town (Jianchuan County, Dali Bai Autonomous Prefecture, Yunnan Province, China) has timber forests, economic forests, charcoal forests, protective forests, and other multi-purpose forests.

Pingbian County is located in the south of Yunnan Province and the southeast of Honghe Prefecture, south of the Tropic of Cancer. It has wet and rainy forests and a forest coverage rate of 68.3%, having the reputation of being a "natural oxygen zone" (Figure 2). The county's minority population accounts for 67.5%, and is the only Miao autonomous county in Yunnan Province, with 44.68% of the county's total population belonging to the Miao people. The case sites were selected in the Baihe Town and Baiyun Town of Pingbian County (Honghe Hani and Yi Autonomous Prefecture, Yunnan Province), where timber forests and ecological public welfare forests account for a relatively large area.

The case sites are located in states with abundant forest resources, and the data of the total forestry output value of each state and city were collated and divided from the 2021 Yunnan Statistical Yearbook. The two states are in the middle to upper class, with sufficient endogenous power for forestry industry development; thus, the study of behavior and willingness of the ethnic minority forest landowners to engage in forest land management is somewhat representative (Figure 3).

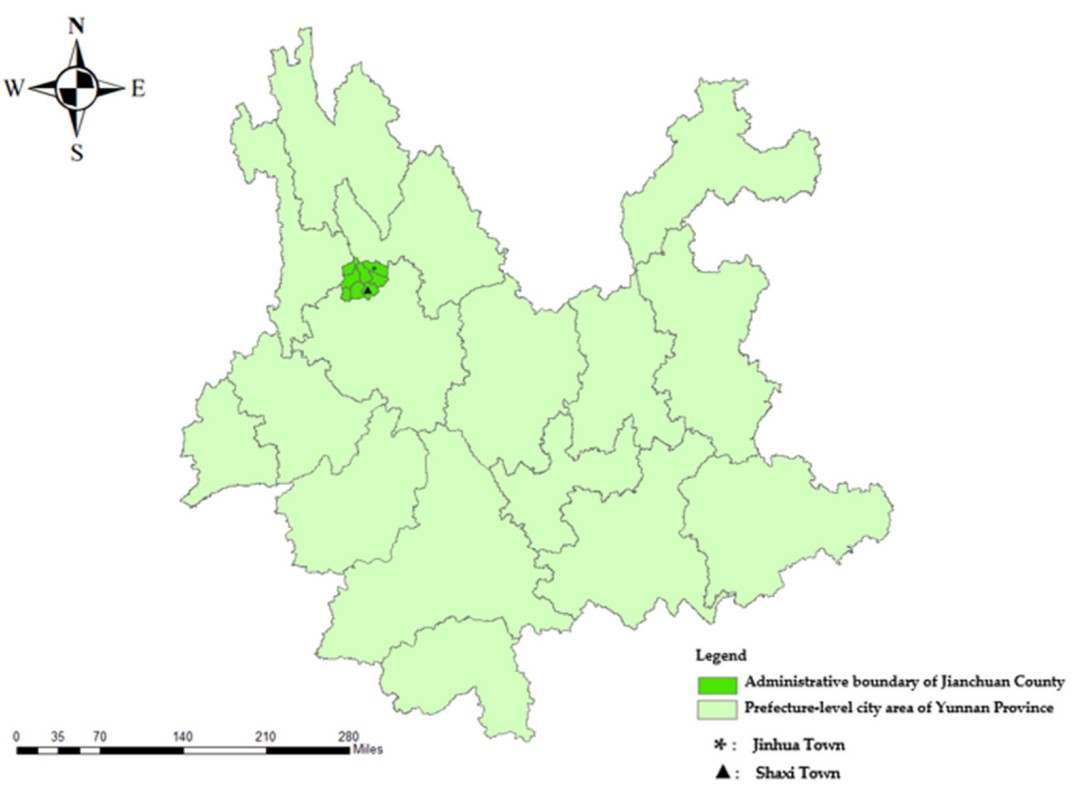

**Figure 1.** Administrative boundary map of Jianchuan County (Dali Bai Autonomous Prefecture, Yunnan Province, China).

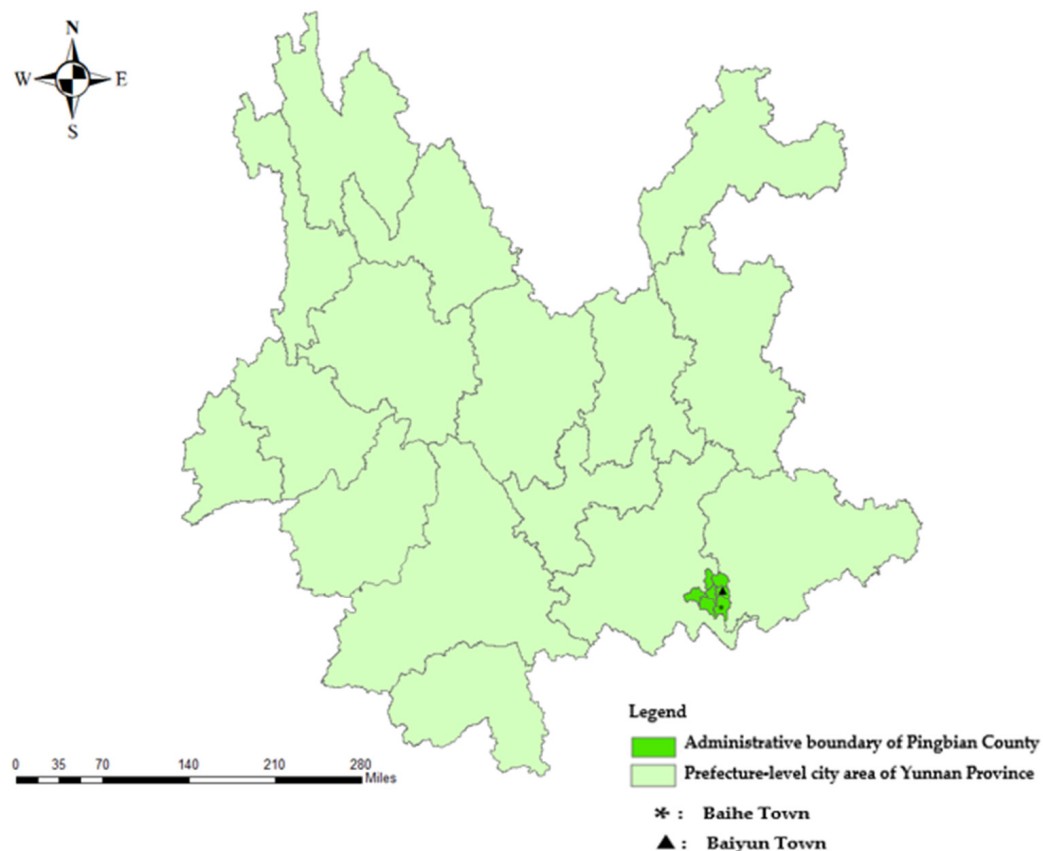

**Figure 2.** Administrative boundary map of Pingbian County (Honghe Hani and Yi Autonomous Prefecture, Yunnan Province, China).

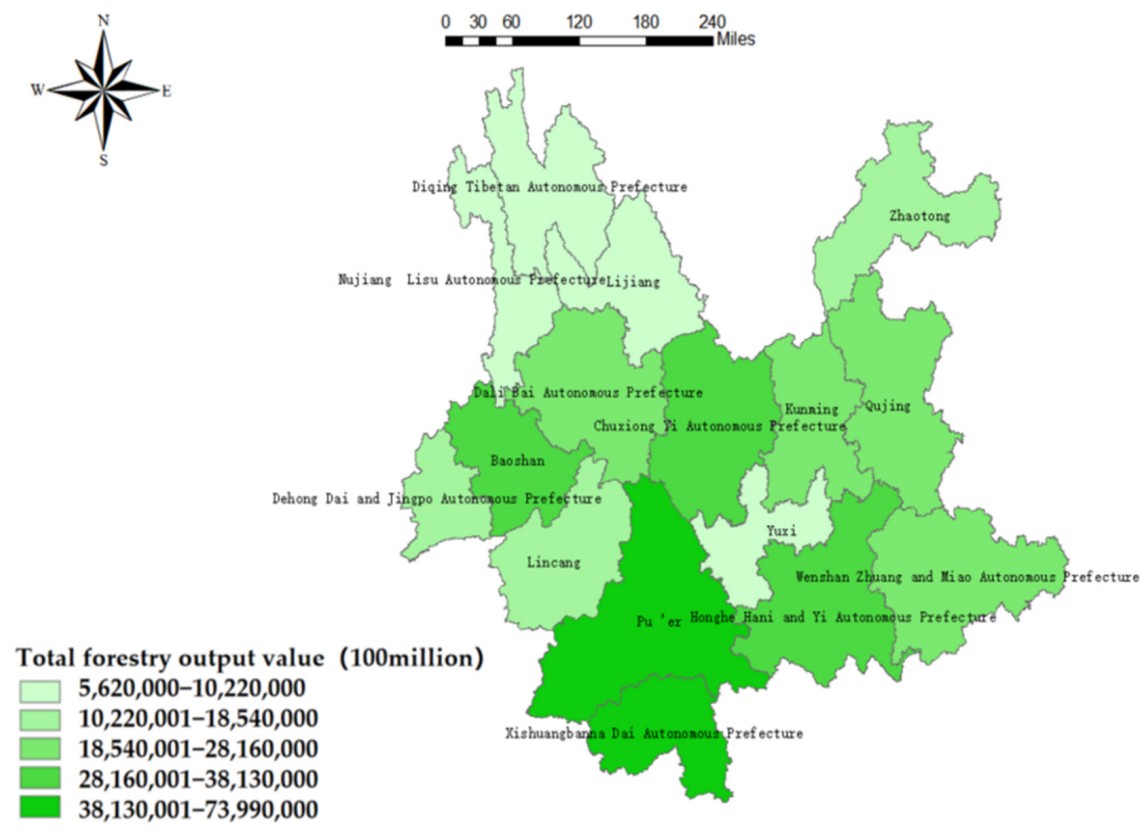

**Figure 3.** Distribution of forestry-output value in Yunnan Province (China).

*2.2. Data Sources*

Since the full implementation of the collective forest rights system reform in Yunnan Province in 2007, a large amount of forest land resources has been allocated to ecological public welfare forests and natural forest protection areas. In order to gain an in-depth understanding of the basic overview of forest land management in the ethnic minority regions of Yunnan after the collective forest rights system reform, we were commissioned by the comprehensive research and evaluation project team of Peking University on 23 August–4 September 2021 to conduct fieldwork. The team mainly asked farmers survey questions in the form of sample surveys. We debugged the equipment and recorded the farmers' answers using an iPad in order to improve the recovery rate and efficiency of the electronic questionnaire. Compared with the traditional paper version of the questionnaire, this avoids the disadvantages of unclear handwriting, data loss and heavy collation tasks. As the foresters answer the questions, the iPad saves the data in a timely manner so that there are no missing data. Our team members also conducted structured surveys with groups, such as elderly villagers, forestry leaders, village cadres, and township leaders, as textual references, which also support and have important significance for our research (Appendix A). The interviews focused on obtaining answers to questions about basic information, the business overview and model, the willingness to manage forest land, and the reform of collective forest rights system. After the survey was completed, the electronic questionnaire saved on the iPad was saved directly in the backend app database, and under the guidance of the project leader, our team members organized the questionnaire. Data about which we were in doubt were reassessed by re-interviewing the surveyed foresters, visiting them again to improve the accuracy and validity of the questionnaire. Therefore, the survey of farmers in this context was comprehensive, and the questionnaires were able to reflect the current living standards of farmers and the development of agricultural and forestry-related activities in various aspects, including their production and life.

The research team was divided into 2 groups, and the investigators were all enrolled through the recruitment system of the collective forest rights system visitors. They were all college students or graduate students majoring in rural development, urban and rural planning, and agricultural and forestry economic management. Among the 5 investigators in Jianchuan County (Dali Bai Autonomous Prefecture, Yunnan Province, China), 3 of them were local people from Jianchuan County, and 1 of the 5 investigators in Pingbian County (Honghe Hani and Yi Autonomous Prefecture, Yunnan Province, China) was local to Pingbian County. Under the leadership of the two counties' forestry and grassland bureaus, the investigators first conducted surveys with county leaders and explained the reason for the survey, followed by village leaders and village group leaders who led the investigators to the survey households. A total of 185 questionnaires were issued, including 101 in Jianchuan County (Dali Bai Autonomous Prefecture, Yunnan Province, China) and 84 in Pingbian County (Honghe Hani and Yi Autonomous Prefecture, Yunnan Province, China). The survey randomly selected a total of 185 ethnic minority households including Bai, Yi, Lisu, and Miao, as survey respondents, of which the number of valid questionnaires was 182, with an efficiency rate of 98.38%. The data of the three questionnaires were still not available as supporting material after the return visit, so they were excluded as invalid questionnaires when the model analysis was conducted.

The field survey involved 185 minority farming households from 8 of the region's minority groups in 10 villages in Jianchuan County (Dali Bai Autonomous Prefecture) and Pingbian County (Honghe Hani and Yi Autonomous Prefecture) of Yunnan Province (Table 1).

**Table 1.** Basic overview of field research sites.

| State | County | Village Name | Number of Minority Households Surveyed | Minorities Surveyed |
|---|---|---|---|---|
| Dali Bai Autonomous Prefecture | Jianchuan County | Jinhua South Gate Community | 19 | Bai, Yi, Lisu, Hui, Naxi |
| | | Jinhua West Gate Community | 23 | |
| | | Aofeng Village of Shaxi Town | 21 | |
| | | Beilong Village of Shaxi Town | 20 | |
| | | Southeast Village of Shaxi Town | 18 | |
| Honghe Hani and Yi Autonomous Prefecture | Pingbian County | Fangyang Village of Baihe Town | 16 | Miao, Yi, Zhuang, Yao |
| | | Mabuchong Village of Baihe Town | 18 | |
| | | Shengli Village of Baihe Town | 17 | |
| | | Taiping Village of Baiyun Town | 15 | |
| | | Baiyun Village of Baiyun Town | 18 | |

Data source: Based on questionnaires.

### 2.3. Variable Selection

The Sustainable Livelihoods Analysis (SLA) framework is a research tool based on the British Development Agency, which has been widely used in the analysis of livelihood vulnerability [44] and poverty [45]. It also considers the influencing factors and willingness to manage from the perspective of farmers, [46] which intuitively reflects the problems and needs at the micro-level. William D. Sunderlin summarized the study of rural livelihoods, such as forest resource protection, utilization, and poverty reduction measures practiced in developing countries [47]. Nimai Das studied the impact of participatory forestry programs on rural livelihood sustainability outcomes in poor households in India [48]. The sustainable livelihood framework is also used as a research tool for the sustainable development of grassland ecosystems [49], the implementation effect of the grassland ecological compensation policy [50], joint forest management [51], and other fields related to forestry ecological development. Therefore, in the existing research, the application of the SLA framework in the field has been more common and extensive. The research areas are mostly concentrated in poverty-stricken and remote minority regions [52]. The research

content mostly focuses on the livelihood status of farmers and the livelihood factors that affect the income level or behavior of farmers. Few studies have been conducted on key ecological functional areas such as Yunnan Province, and there are few studies on the influencing factors of the SLA framework for the willingness of ethnic minority forest landowners to engage in forest management.

Forest land resources are the key production factors for landowners in ethnic minority regions to develop the forestry industry. Forest landowners in ethnic minority regions are influenced by the traditional behaviors of their ancestors, from the primitive tribal group lifestyle to the traditional smallholder management model and modern management, and their production and living behaviors such as hunting and harvesting, food customs, and firewood collection, are all involved in forest land management in a direct or indirect way, showing the high degree of dependence on forest resources by farmers in ethnic minority regions. This paper drew on the human capital, natural capital, physical capital, financial capital, and social capital from the Sustainable Livelihood Analysis framework (SLA) [53,54], which can visually reflect the strengths or weaknesses of forest landowners in ethnic minority regions in terms of the various capital elements in forest land management (Figure 4).

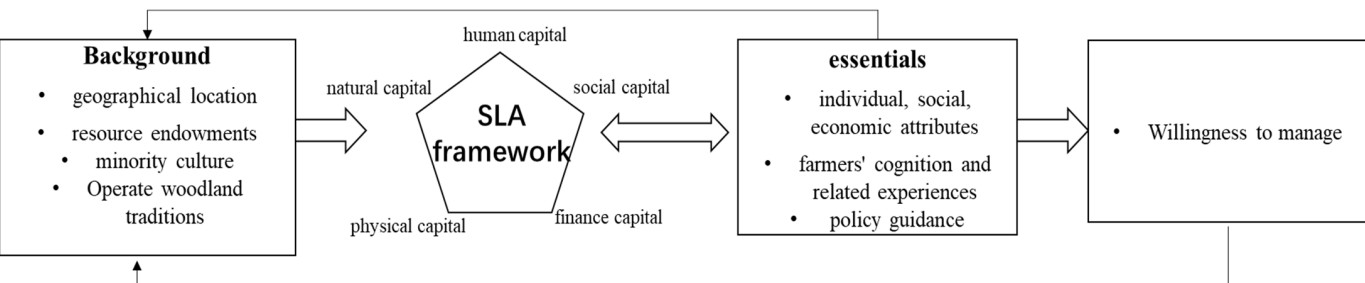

**Figure 4.** Framework diagram of Sustainable Livelihood Analysis.

Livelihood capital is a prerequisite for foresters to choose livelihood strategies and execute livelihood behaviors. The farmers' willingness to manage forest resources comes from their rational decisions, and their perceptions and related experiences as well as variables of individual social economic attributes, and policies are important elements to be considered in their rational decisions. In order to understand more intuitively the impact of the five categories of capital in the SLA framework on the foresters' willingness to manage forests in ethnic minority areas, as shown in Figure 5, we regrouped the five categories of capital with reference to the existing studies [54]. (1) The only variable included in human capital is the *education level*. (2) Social capital includes variables such as *forest landowner identity*; *the understanding of scale operations*; *participation in joint account operation*; *participation in the project of returning farmland to forest and grassland*; and *the binding force of the harvesting quota policy, scored on a scale of 1–10*. (3) Financial capital includes *whether they have been compensated by public welfare forests*. (4) The variable included in physical capital is the *standard of living*. (5) The variables included in natural capital are *woodland area*; *woodland feature type*; and *whether they are satisfied with the implementation of the reform policies*.

Therefore, according to the SLA framework and combined with the field survey results, we used the following factors: forest landowners' identity, education level, living standard, forest land area, and forest land function type as indicators of the influence of the forest landowners' individual level, economic level, and social level on the willingness to manage forests. These determine the forest landowners' management efficiency and perception of management risk. At the same time, the perceptions of joint-family operation and large-scale management, and the participation in the project of returning farmland to forests and grasses were also used as variables influencing the forest landowners' perceptions and related experience levels regarding management intentions. In addition, the three variables of compensation for public welfare forests, satisfaction with the collective forest rights

system reform, and the strength of harvesting limit policy constraints also reflect whether policy guidance has a subjective-level effect on the ethnic minority forest landowners' management intentions.

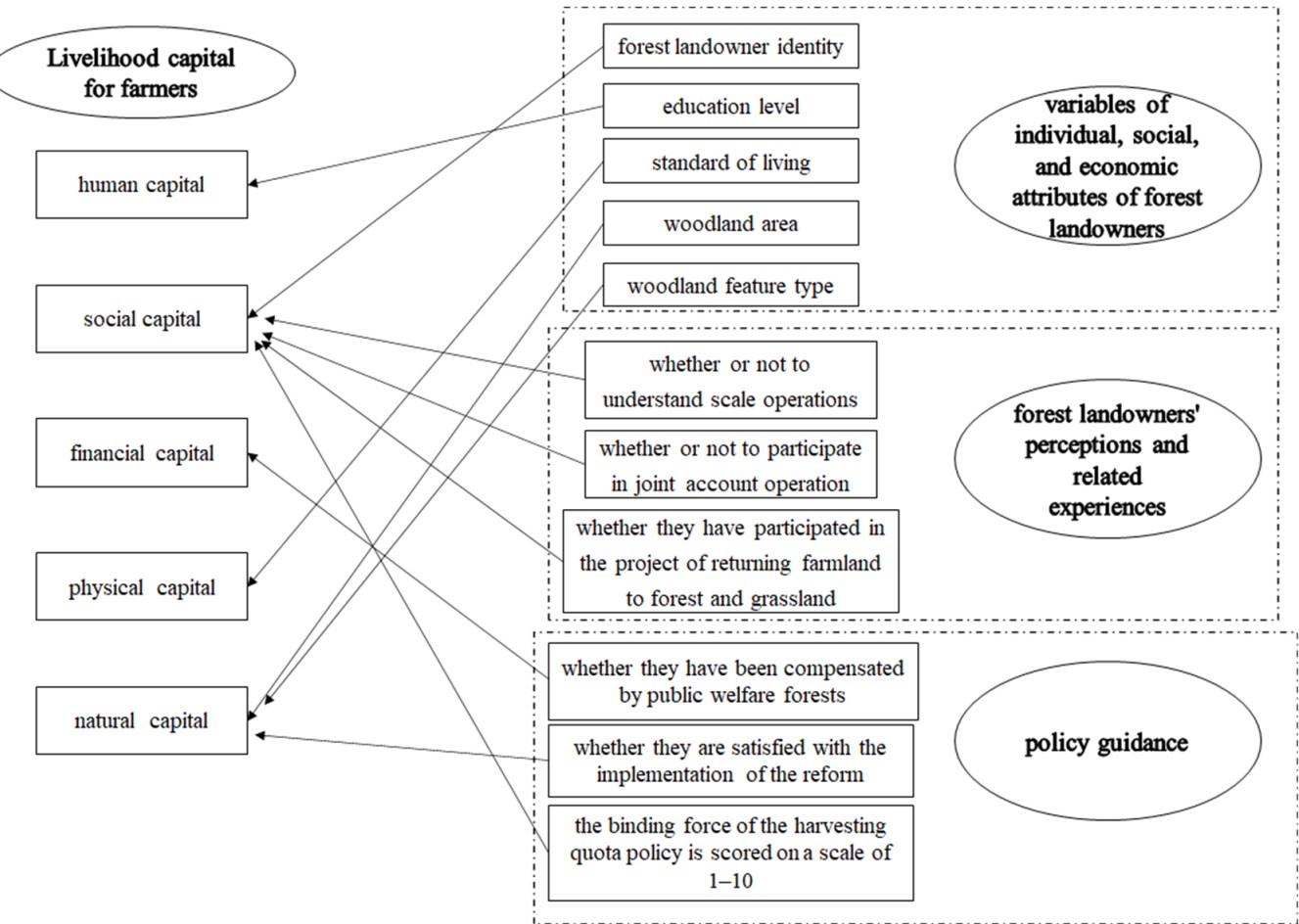

**Figure 5.** Mechanistic analysis framework of the impact of livelihood capital on variables.

Additionally, there are 3 issues to consider regarding forest landowners in ethnic minority regions [55]: First, forest landowners in ethnic minority regions may have stronger ethnic minority plots, and their deep-rooted beliefs form a self-protection mechanism compared to the management behavior and willingness of forest landowners in non-ethnic minority regions. Therefore, even if they do not manage forest land, they will not change their behavior and are more willing to possess forest land resources in their own hands or pass them on to the next generation. Second, based on the geographical characteristics and resource endowment of ethnic minority regions, the distance between forest plots and the problem of fragmentation are more prominent, which affects the forest revenue and the implementation of mechanized operations. The forest revenue directly affects the management behavior and thus the willingness of farmers to engage in forest management, and there are always some obstacles to changes in the behavior and willingness to manage forests. Third, the proportion of ecological public welfare forests in Yunnan's ethnic minority regions is high, and the harvesting target and quota policy greatly limit the forest landowners' enthusiasm and their willingness of manage it.

In summary, three research hypotheses are proposed.

**Hypothesis 1 (H1).** *Variables of individual social economic attributes of forest landowners are conducive to strengthening the forest landowners' willingness to engage in forest management.*

**Hypothesis 2 (H2).** *Forest landowners' perceptions and related experiences are conducive to strengthening the forest landowners' willingness to engage in forest management.*

**Hypothesis 3 (H3).** *Policy guidance is conducive to strengthening the forest landowners' willingness to engage in forest management.*

The dependent variable is the willingness to engage in forest management, and this dependent variable can be indexed subjectively to relevant studies [56–60]. It was found that the subjective variables reflect the future behavior or preferences of individuals, and that the answers of the studies differed from one respondent to another and from one geographical area to another [61] and were somewhat comparable.

As shown in Table 2, as far as the factors of individual, social and economic attributes of farmers are concerned, based on the human and social capital considerations in the Sustainable Livelihood Analysis (SLA) framework, the influence of these two types of capital on the efficiency of forest land use and willingness to manage is mainly reflected in the variables of farmers' identity, education level, living standard, woodland area [62], and functional type of forest land. This will constrain the farmers' forest land management efficiency, future management risk, and the judgment of management philosophy, based on their education levels, will affect the adoption behavior of production tools and advanced technology. In terms of the factors that affect the forest landowners' perceptions and related experiences, forest landowners judge the ease of future management and the benefits based on their perceptions of the management mode [63] and their actual participation in forestry-related projects in the past [64], so as to assess whether it is worthwhile to continue this management in the future. Therefore, three variables including the perceptions of joint-family operation and large-scale operations, and the participation of returning farmland to forestry and grass, were summarized. In terms of the influencing factors at the policy guidance level, social capital considerations based on the Sustainable Livelihood Analysis (SLA) framework can subjectively regulate, and have an incentive effect on, the forest landowners' willingness to manage forests [65,66], including the three variables of being subject to compensation for public welfare forests, satisfaction with the collective forest rights system reform, and the evaluation of harvesting quota policy constraints.

**Table 2.** Descriptive statistics of factors influencing willingness to engage in forest management.

| Variable Type | Variable Name (Code) | Definition and Assignment | Average Value | Standard Deviation |
|---|---|---|---|---|
| dependent variable | *willingness to engage in forest management* ($Y$) | Yes = 1; No = 0 | 0.720 | 0.449 |
| variables of individual, social, and economic attributes of forest landowners | *forest landowner identity* ($X_1$) | cadres = 1; ordinary people = 0 | 0.423 | 0.495 |
| | *education level* ($X_2$) | read more = 1; read less = 0 | 0.170 | 0.377 |
| | *standard of living* ($X_3$) | poverty = 1; wealthy = 0 | 0.291 | 0.456 |
| | *woodland area* ($X_4$) | actual value (hectare) | 0.9226 | 0.583 |
| | *woodland feature type* ($X_5$) | commercial forest = 1; ecological public welfare forest = 0 | 0.736 | 0.442 |
| forest landowners' perceptions and related experiences | *understand scale operations* ($X_6$) | Yes = 1; No = 0 | 0.093 | 0.292 |
| | *participate in joint account operation* ($X_7$) | Yes = 1; No = 0 | 0.088 | 0.284 |
| | *participated in the project of returning farmland to forest and grassland* ($X_8$) | Yes = 1; No = 0 | 0.462 | 0.500 |

**Table 2.** *Cont.*

| Variable Type | Variable Name (Code) | Definition and Assignment | Average Value | Standard Deviation |
|---|---|---|---|---|
| policy guidance | *whether they have been compensated by public welfare forests* ($X_9$) | Yes = 1; No = 0 | 0.670 | 0.471 |
| | *whether they are satisfied with the implementation of the reform* ($X_{10}$) | Yes = 1; No = 0 | 0.857 | 0.351 |
| | *the binding force of the harvesting quota policy is scored on a scale of 1–10* ($X_{11}$) | actual value | 8.533 | 1.627 |

*2.4. Data Analysis*

The models that were used more often in the empirical analysis of the factors that influence farmers' willingness to manage in the existing studies are the Probit regression model [67], structural equation model [59], and logistic regression model [68]. Considering the existing research and combined with the questionnaire data, the logistic regression model was selected for the empirical analysis in this paper. In addition, the *Y* variable in this paper was a dichotomous variable, so the binary logistic regression model in the logistic regression model was used to analyze the influencing factors of different levels of forest landowners' willingness to engage in forest management in ethnic minority regions.

First, we processed the data. In the study of forest landowners' willingness to engage in forest management in ethnic minority regions, forest landowners' identity, scale of operation, and living standard are all influencing factors, and variables such as *forest landowner identity* and *standard of living* belong to a fixed category of data. Therefore, virtual variable processing was conducted. Taking "*forest landowner identity*" as an example, the answer "cadres" was assigned the value of 1, and "ordinary people" was assigned 0.

Secondly, after completing the above data processing, the *Y* variable was encoded. Forest landowners in ethnic minority regions have two options of "willing" and "unwilling" for forest land management, which is a binary variable and a typical binary selection model. We assign a value of 1 to the willingness to engage in forest management and 0 to the nonwillingness to manage it. It was assumed that the error term obeys the logistic distribution.

Finally, the analysis of the influence relationship, binary logistic regression analysis, was performed. We first determined whether an influence factor appears to be significant (if the *p*-value is less than 0.05, then it is significant at the 0.05 level), and if it appears to be significant, the independent variable has an influential relationship on the dependent variable of the willingness to engage in forest management. After determining the influence relationship, the analysis was conducted in conjunction with the regression coefficient value; if the regression coefficient value is greater than 0, the influence relationship is positive, and vice versa, it is negative.

The binary logistic model equation is as follows:

$$ln\left[\frac{P(Y=1)}{1-P(Y=1)}\right] = \alpha + \beta_i X_i + \varepsilon \tag{1}$$

where $P(Y=1)$ represents the forest landowners' willingness to engage in forest management; $P(Y=0)$ represents the forest landowners' lack of willingness to engage in forest management; $X_i$ denotes the *i*th influencing factor; $\alpha$ is a constant term; $\beta_i$ is an estimated parameter; $\varepsilon$ denotes a random variable obeying logistic distribution, and $P(Y=1 \mid X_1, X_2, X_3 \cdots X_i)$ is the probability that forest landowners in ethnic minority regions are willing to engage in forest management under the influence of *i* independent variables.

## 3. Results

### 3.1. Model Regression Results and Tests

As shown in Table 3, the model was evaluated for its validity and the overall model fit was likelihood = 158.414, $p$ = 0.000. It was significant at the level and rejected the original hypothesis, thus indicating that the model fit was good and valid overall. The classification effect of the logistic regression can be measured in the evaluation results of classification indexes, where the value of accuracy is 0.808, which predicts the proportion of correct samples to the total samples. The closer the value is to 1, the higher the number of correct samples in the model classification evaluation indexes. The value of F1 reflects the reconciled average of accuracy and recall of the survey data; its value is 0.794, which is a good effect; the value of the AUC value is 0.836, which is closer to 1, indicating the better classification effect of the indicators, which also proves that the classification of factors that influence the forest landowners' willingness to engage in forest management in ethnic minority regions according to the sustainable livelihood framework is consistent with the model regression.

**Table 3.** Model evaluation results.

| Likelihood Ratio Chi-Squared Value | $p$ | Sample Accuracy | F1 | AUC |
|---|---|---|---|---|
| 158.414 | 0.000 *** | 0.808 | 0.794 | 0.836 |

Note: *** represents 1% level of significance.

The model regression results are shown in Table 4.

**Table 4.** Model regression results of forest management intention.

| Argument | Regression Coefficient | Standard Error | Salience |
|---|---|---|---|
| *forest landowner identity* ($X_1$) | −0.788 | 0.591 | 0.183 |
| *education level* ($X_2$) | −2.21 | 0.807 | 0.006 *** |
| *standard of living* ($X_3$) | 2.359 | 0.715 | 0.001 *** |
| *woodland area* ($X_4$) | 0.095 | 0.034 | 0.006 *** |
| *woodland feature type* ($X_5$) | −0.229 | 0.461 | 0.619 |
| *understand scale operations* ($X_6$) | −0.055 | 0.644 | 0.932 |
| *participate in joint account operation* ($X_7$) | −0.108 | 0.703 | 0.878 |
| *whether they have participated in the project of returning farmland to forest and grassland* ($X_8$) | −1.48 | 0.441 | 0.001 *** |
| *whether they have been compensated by public welfare forests* ($X_9$) | −1.25 | 0.446 | 0.005 *** |
| *whether they are satisfied with the implementation of the reform* ($X_{10}$) | 0.446 | 0.635 | 0.483 |
| *the binding force of the harvesting quota policy is scored on a scale of 1–10* ($X_{11}$) | 0.212 | 0.146 | 0.147 |

Note: *** represents a significance level of 1%.

### 3.2. Effectiveness of the Collective Forest Rights Reform and Forest Landowners' Willingness to Engage in Forest Management

(1) The reform of the collective forest rights system in Yunnan Province is effective. According to the questionnaire data, regarding the number of plots owned by forest landowners in the case sites, the percentage of plots with forest land right certificates is 93.53%, among which 64.8% of forest landowners in ethnic minority regions said they obtained forest land right certificates in 2007. This was at the stage of the comprehensive reform of the collective forest rights system in Yunnan Province, where clear property rights provide rights protection for activities such as forest land management, adjustment of disputes, and application for logging. On the one hand, empowering rural communities or farmers to transform forest land into households is not only conducive to solving the

problem of asymmetry between forest use rights and protection responsibilities. On the other hand, farmers can achieve the goal of increasing forest income services for farmers and contributing to sustainable forest management through forest land leasing, transfer, and contracting. There were very few forest landowners (5.59%) who used forest right certificates to mortgage loans to obtain operating capital, which indicates the special behavior and conservative attitude of forest landowners in ethnic minority regions towards mortgage risk and operating financing (Table 5). This is very different from the large-scale circulation and rental of forest land by farmers in non-ethnic minority areas.

**Table 5.** Descriptive statistics table of woodland rights confirmation.

| Overview of Woodland Rights Confirmation | Number of Plots (Blocks) | Percentage (%) | Whether or Not to Obtain a Loan | Number of Plots (Blocks) | Percentage (%) |
|---|---|---|---|---|---|
| have a woodland title certificate | 289 | 93.53 | Yes | 17 | 5.59 |
| | | | No | 287 | 94.41 |
| no woodland title certificate | 20 | 6.47 | \ | | |

Data source: Based on questionnaires.

(2) The forest landowners' willingness to produce and manage forests is strong. Among the 182 valid questionnaires collected, 131 minority forest landowners were willing to engage in forest management, accounting for 71.98%; 51 minority forest landowners were not willing to engage in forest management, accounting for 28.02%. The main reasons for their unwillingness to engage in forest management were the low subsidy standard for public welfare forests and policy restrictions, and the high proportion of no-harvesting targets, at 41.18% and 27.45%, respectively (Table 6). Some forest landowners responded that they were unwilling to continue to manage their forest land due to low income, the distance of the forest land, and the inconvenience of management.

**Table 6.** Descriptive statistics table of willingness to manage and reason.

| Whether There Is a Willingness to Engage in Forest Management | Frequency | Percentage (%) | Reason | Frequency | Percentage (%) |
|---|---|---|---|---|---|
| No | 51 | 28.02 | low yield | 10 | 19.61 |
| | | | the subsidy standard for public welfare forests is too low | 21 | 41.18 |
| | | | the woodland is finely fragmented and far away | 6 | 11.76 |
| | | | policy restrictions; no logging indicators | 14 | 27.45 |
| Yes | 131 | 71.98 | \ | | |

Data source: Based on questionnaires.

## 4. Discussion

This paper addressed the research gap of micro-level studies related to ethnic minority areas and summarized the reasons that influence foresters' willingness to manage forests in ethnic minority areas. We addressed the micro-level perspective to explain the reasons why farmers in ethnic minority regions are influenced by different levels of philosophy, endowment, and policy in their management of forest land, which lead to different results from those of studies in non-ethnic minority regions.

*4.1. Influence of "the Variables of Individual Social Economic Attributes of Forest Landowners" on Their Willingness to Engage in Forest Management*

(1) *Education level.* The regression results showed a negative effect of "more education" and "less education" on the foresters' willingness to manage forests. The regression results

show that *education level* has a negative effect at the 1% significance level, indicating that the more educated the forest landowners in ethnic minority regions are, the less willing they are to manage forest land, which is inconsistent with the hypothesis. Rong Niu also found a negative effect of literacy in his survey on the willingness of creditors to lend the farmland management rights in the western region [69]. The reasons for this may include the following. First, the higher the education level, the easier it is for forest landowners to obtain stable, well-paid jobs, and the more willing they are to work outside the home to sustain their livelihoods, with higher opportunity and sunk costs of giving up their current positions [70]. Secondly, forest landowners with relatively less education have an earlier access to agricultural production and management activities and become the main force of forest land management. Compared with forest landowners with more education and schooling, their sentimental attachment to the land is deeper, and they have fewer channels to obtain other sources of income and less sensitive information [71], which directly affects their behavioral attitude and willingness to manage forests.

(2) *Standard of living*. The effect of *standard of living* on the willingness to engage in forest management was measured by the two options of "poor" and "rich". The regression results show that the *standard of living* have a positive effect at the 1% significance level; forest landowners with relatively lower living standard are more willing to engage in forest management. In a study on multidimensional poverty in rural Bihar, India, Manjisha Sinha found that forest landowners with a higher dependence on livelihood activities such as forestry and higher poverty levels were more affected by climate change and their business behavior was considerably more influenced by objective factors [72]. In contrast to the behavior of forest landowners in these areas lacking the characteristic of the ethnic minority regions, the willingness of farmers in ethnic minority regions to manage forests is instead more influenced by subjective factors. Possible reasons for this include the following. First, forest landowners with relatively low living standards have a weaker ability to bear risks, and agricultural production and management activities are less risky compared to other industries; forest landowners with relatively high living standards have a certain financial ability and prefer investment-oriented activities, and their behavioral attitudes make their willingness to engage in forest management in the future smaller [73]. Secondly, the long payback period of forest land investment, low income and low efficiency of production, and the special nature of operation mean that income cannot meet people's living needs [74].

(3) *Woodland area*. The actual value of the *woodland area* was used as an economic attribute variable to analyze the effect on the forest landowners' willingness to manage forests in terms of hectare. The regression results show that the *woodland area* has a positive effect at the 1% significance level; forest landowners with more woodland area are more willing to continue operating their woodlands. The per capita forest land area of the case sites reached 0.9226 hectares, which just confirms that forest landowners in ethnic minority regions have been at the center of the forest-centered ecosystem for a long time and are more sentimentally attached to the land, and the endowment of forest resources in ethnic minority regions provides an important material basis for farmers to maintain their livelihoods. Despite the ineffectiveness of forest land management, farmers who own more forest land are more willing to maintain their resources in their own hands and have greater expectations of forest land, while compared to some forestry households in Korea, the magnitude and direction of the impact of different acreage on different income types are inconsistent [20], which is the difference in behavioral attitudes of farmers in ethnic minority regions compared to farmers in non-ethnic minority regions in terms of forest land management behavior and willingness.

### 4.2. Influence of "Forest Landowners' Perceptions and Related Experiences" on the Willingness to Engage in Forest Management

Among the three variables of forest landowners' perceptions and related experiences, most of the forest landowners in the studied ethnic minority regions did not participate in the large-scale and joint-family management of forest land, so they did not know much

about the concept, and thus these two variables did not have a significant effect on their willingness to engage in forest engagement. Only "*whether they have participated in the project of returning farmland to forest and grassland*" had a negative effect on the willingness to engage in forest management at the 1% level of significance, which is not consistent with the expected hypothesis. However, the respondents were more satisfied with the policy of "returning farmland to forest and grass", while the willingness of farmers in non-ethnic minority regions was different [75]. It may be due to the fact that most of the forest landowners used to cultivate food and burn and cut firewood, but they had to change their cultivation habits due to policy restrictions after the implementation of the project; secondly, the basic and non-basic farmland are intertwined in most of the ethnic minority regions in Yunnan, which makes the forest land more fragmented and more difficult to manage, and the subsidies for returning farmland to forest and grass are not proportional to their expectations. This has a direct impact on the attitude of forest landowners, which leads to a low willingness of forest landowners to engage in forest management.

In addition, based on the content of the interviews, it was found that the forest culture of different ethnic minorities has special characteristics, which are concentrated in nature worship, religious belief, and totem culture. They conduct different forest protection activities, ethnic traditions and customs concentrated on the reverence for mountain gods and tree gods, and the concept of animism [76]. Although forest farmers in ethnic minority areas do not manage forest land in a direct way, forest resources have varying degrees of importance and influence on them. Forest farmers in non-ethnic minority areas have almost no beliefs and cultural practices concerned with the management of forest land, and due to factors such as geography and resource conditions, the scale and continuous management mode are relatively common and the cost is low; thus, most of them manage forest land in a direct way. It can be seen that there are great differences in the behavior, philosophy, and willingness of forest farmers to manage forests in ethnic minority areas and those in non-ethnic minority areas.

*4.3. Influence of "Policy Guidance" on the Willingness to Engage in Forest Management*

Among the three variables of policy guidance, "*whether they have been compensated by public welfare forest*" had a significant negative impact on the willingness to manage forest land, which is inconsistent with our expectations. This is an important phenomenon regarding the factors that affect forest landowners' willingness to manage forests in ethnic minority regions. Generally speaking, compensation tends to increase farmers' willingness to engage in forest management [77]. The reasons for this may be as follows. First, compared to non-ethnic minority areas, the proportion of ecological public welfare forests in ethnic minority regions is high, forest landowners are more restricted by the logging quota policy and the comprehensive ban on natural forests, and they are not fully knowledgeable about institutional policy constraints [78]. With the increase in ecological awareness, the foresters' enthusiasm to manage the land is greatly frustrated; thus, forest landowners prefer to maintain ownership of the forest land use and the use rights. Secondly, during the process of the interview, forest landowners, management entities, and village leaders provided feedback on the low compensation standards of ecological public welfare forests, reflecting the fact that the subjective norms of the policy directly affect forest landowners' own economic rationality, thus leading to the emergence of different degrees of management willingness.

In other scholars' studies, it was found that, in terms of geographical location conditions, the more developed areas in east-central China, areas with good forestry resource endowment, and areas with significant reform policies are relatively more efficient in developing the forestry industry and more effective in large-scale management [23]. Academic research is also focused on the relevant regions with more diversified business models and where farmers are willing to engage in forest management, as concluded by Han [79], Zheng [24], and Hu [80]. However, in this study, it was found that the factors of scale

operation and diversification have no effect on the willingness of ethnic minority farmers to engage in forest management.

*4.4. Research Shortcomings and Outlook*

Through the empirical analysis of the willingness of farmers in ethnic minority regions to manage forest land in the Jianchuan County (Dali Bai Autonomous Prefecture, Yunnan Province, China) and Pingbian County (Honghe Hani and Yi Autonomous Prefecture, Yunnan Province, China), we found shortcomings in the forest landowners in ethnic minority regions regarding their forest management ideology and behavioral characteristics. The innovation of this paper was its adoption of the framework of sustainable livelihood analysis from a sociological perspective as the theoretical support. We explained the unique trends in the willingness of forest landowners to manage forests in ethnic minority regions from different perspectives, such as ethnology and ecology, and obtained conclusions that were identical to other academic studies and reflect differences from non-ethnic minority regions in terms of the research results and the direction of the influence of factors on the willingness to manage forests.

There are some limitations to this study. There are twenty-five minority groups living in Yunnan Province, and this study only analyzed eight of them, thus lacking research on groups in other minority regions. Secondly, our team members went to the Jianchuan County (Dali Bai Autonomous Prefecture, Yunnan Province, China) and Pingbian County (Honghe Hani and Yi Autonomous Prefecture, Yunnan Province, China) to conduct a comprehensive investigation and evaluation of Peking University's collective forest tenure system reform. Since the questionnaire design mainly focused on the content of the reform, and we used in this article the relevant information extracted from the questionnaire, the selected variables were limited, and the results of the specific measurement of the factors that influence the forest landowners' willingness to manage forests in ethnic minority regions should vary according to the actual location and variables. Therefore, future studies should focus on the dynamic follow-up of the factors that influence the forest landowners' willingness to engage in forest management in ethnic minority regions at the specific minority and micro levels.

## 5. Conclusions

A study was conducted on ethnic minority farming households in ethnic minority regions in the Jianchuan County (Dali Bai Autonomous Prefecture, Yunnan Province, China) and Pingbian County (Honghe Hani and Yi Autonomous Prefecture, Yunnan Province, China). A binary logistic model was used to empirically analyze the effects of three-dimensional variables, namely, variables of the individual social economic attributes of forest landowners, the cognitive and related experiences of farming households, and policy guidance, on their willingness to engage in forest management. The results of the study show that *standard of living* and *woodland area* have a significant positive influence on the willingness to engage in forest management, whereas *education level, whether they have participated in the project of returning farmland to forest and grassland*, and *whether they have been compensated by public welfare forests* have a significant negative influence on the willingness to engage in forest management. Compared with the factors that influence the willingness of forest landowners to manage forests in non-ethnic minority regions, the possible reasons for this were concluded to be resource endowment, the sentiment of ethnic minority groups for the land, historical habits, and beliefs in ecological forestry concepts.

Firstly, the impact of the variables of individual social economic attributes of the forest landowners' willingness to engage in forest management is relatively significant. According to the above findings, it can be seen that the higher the *education level*, the lower the willingness of the farmers to manage the forests, and the lower the *education level*, the more likely it is that they display business behavior. It was also found that the lower the *standard of living*, the more willing foresters were to manage their forest land. Therefore, the government should promote large-scale forest land management to improve the efficiency

of forest land management and strengthen the collective economy, which can be led by the government through technical training and technical guidance. On the one hand, this is important to improve the added value of forest products; on the other hand, it can enhance the scientific and technological awareness of less educated foresters, reduce their wait-and-see attitude, change the concept of foresters, and improve the overall knowledge level of foresters.

Regarding the conclusion that forest land area positively contributes to the foresters' willingness to manage it, it is argued that foresters in ethnic minority areas have more land sentiment and forest land as natural capital directly affects the livelihood capital of farmers in ethnic minority areas. Therefore, the traditional individual management model can be changed, and forest land can be leased and contracted to large forestry households, management entities, and cooperatives to reduce the cost and effort of individual-scale management. Thus, the advantages of forest resources in ethnic areas can be brought into play more effectively and the value of natural capital can be preserved and increased.

Secondly, forest landowners' cognition and related experiences have a significant negative impact on their willingness to engage in forest management "*whether they have participated in the project of returning farmland to forest and grassland*". Considering the special resource endowment, the landscape characteristics, and the ethnic minority groups' sentimental affection for the land, the government can help by extending the subsidy period and increasing the subsidy standard, strengthening scientific and standardized management according to the conditions of different fallow land plots. Since forest landowners in minority regions have a very low understanding of large-scale operation and joint-family operations, the government should reasonably adjust the plots to achieve concentrated and large-scale operations, which will also break through the limitation of forest land fragmentation.

Thirdly, the results of the analysis of the impact of public welfare forest compensation standards on the willingness of forest land management show that due to the large proportion of ecological public welfare forests in ethnic areas and limited financial resources, many ethnic minority foresters are dissatisfied with the current compensation standards, which also indicates that the current policies introduced in relation to public welfare forest compensation do not meet the needs of ethnic minority foresters. Based on the perspective of farmers as "rational economic agents", the compensation rate as financial capital directly influences farmers' livelihood activities, thus affecting their willingness to manage and supply services to the forest system. It is recommended to solve the problems of small compensation scope and low compensation standards, and to implement the differentiated compensation policy of ecological public welfare forest zoning in ethnic minority areas. Reasonable arrangements to stop commercial logging subsidies and incentives for natural forests in ethnic minority areas are directed toward nature reserves, and other woodlands in nature reserves that are not included in the management of public welfare forests are included in the management of public welfare forests.

**Author Contributions:** Conceptualization, H.C. and Y.L.; methodology, H.C.; validation, H.C. and Y.D.; model analysis, H.C.; investigation, H.C. and Y.L.; resources, Y.L.; data curation, H.C.; writing—original draft preparation, Y.L., H.C. and Y.D.; writing—review and editing, Y.D. and X.Z.; supervision, Y.D. and X.Z. All authors have read and agreed to the published version of the manuscript.

**Funding:** This study was funded by: Yunnan Provincial High–level Talent Training Support Program "Youth Top–notch Talent" Special Project. The grant number is: XDYC--QNRC–2022–0427.

**Informed Consent Statement:** This article does not contain any studies with human participants or animals performed by any authors. Informed consent was obtained from all the individual participants included in the study.

**Data Availability Statement:** The data presented in this study are available on request from the corresponding author.

**Acknowledgments:** All authors gratefully acknowledge the support of the people's governments of Jianchuan County (Dali Bai Autonomous Prefecture, Yunnan Province, China) and Pingbian County (Honghe Hani and Yi Autonomous Prefecture, Yunnan Province, China) village leaders, business entities and minority regions' forest landowners for their support of this survey. In particular, we are very grateful to Peking University for funding and guiding the comprehensive survey and assessment of collective forest rights system reform in Yunnan Province.

**Conflicts of Interest:** The authors declare no conflict of interest.

## Appendix A

1. Outline of Interviews with elderly villagers:
   (1) Basic information: ethnic categories; ethnic sentiments; ethnic culture; ethnic customs; ethnic beliefs;
   (2) Willingness to manage forest land: attitude to operation; difficulties in the process of operation; compensation standards for public welfare forests; knowledge and evaluation of the reform of collective forest rights system.

2. Outline of Interviews with large-scale forestry households
   (1) Basic information: forest land resources (forest land area, distribution, etc.); operation cycle; labor use; degree of support in terms of social capital and financial capital;
   (2) Operation profile: forest land transfer and contracting; operation mode (large-scale operation, joint-family operation); mortgage of forest rights; willingness to operate; yield and output value.

3. Outline of Interviews with village cadres, township leaders, forestry bureau department personnel
   (1) Overview of forest land resources in ethnic areas (plot size, distribution distance, forest land function, forest land management);
   (2) Effectiveness and problems of collective forest rights system reform;
   (3) Forestry ecological conservation (conservation effectiveness, ecological awareness);
   (4) Forestry industry development (business cycle, economic benefits, social benefits, ecological benefits).

## Appendix B

Questionnaire on farmers' willingness to manage forest land in ethnic areas after the reform of collective forest rights system.

(I) Basic information
1. Your area: Province, State, County, Town, Village
2. Are you a head of household?
A. Yes B. No
3. Your gender:
A. Male B. Female
4. Your ethnicity is:
5. How old are you?
A. Under 20 years old B. 20–40 years old
C. 40–60 years old D. Over 60 years old
6. What is your education level?
A. Junior high school and below B. High school
C. Specialist D. Bachelor's degree E. Master's degree and above
7. What is your occupation or status?
A. Cadres B. Ordinary people
8. What do you think of your family's standard of living?
A. Poverty B. Wealthy

(II) Forest land resources and the effectiveness of reform

9. How many hectares of forest land do you own: (hectares).

10. What is the type of function of the forest land you own?

A. Commercial forest B. Ecological public welfare forest

11. What is the area of commercial forest you own: (hectares).

12. What is the area of ecological public welfare forest you own: (hectares).

13. How far is the woodland from your home: (km)

14. Do you have a forest land right certificate?

A. Yes B. No

15. Are you satisfied with the effectiveness of the reform of collective forest rights system?

A. Yes B. No

16. Are you satisfied with how the relevant policies have been implemented after the reform?

A. Yes B. No

(III) Factors influencing foresters' experience and willingness to manage

17. Do you have the will to manage forest land?

A. Yes B. No

18. Based on the answers answered in question 17, choose to answer either (1) or (2):

(1) If you have the will to run a business, what is the reason.

(2) If you do not have the will to run a business, what is the reason.

19. Are you involved in joint-family business?

A. Yes B. No

20. Do you understand the scale of business?

A. Yes B. No

21. Have you ever participated in the project of returning farmland to forest and grass?

A. Yes B. No

22. Do you have forest land transfer behavior?

A. Yes B. No

23. Have you tried to obtain a financial loan by pledging a forest land title or management right certificate?

A. Yes B. No

24. In the process of operating forest land, have you been compensated by the public welfare forest?

A. Yes B. No

25. Are you satisfied with the harvesting quota policy for forest land resources?

A. Yes B. No

26. Where do you obtain the bamboo forest harvesting indicators in the course of your business?

A. Village allocation indicators B. Township C. Forestry station

D. Foresters E. Other forestry departments F. Not sure

27. Please rate the logging limit policy on a scale of 1-10 based on your own experience and knowledge: (points).

28. What problems have you encountered in the process of operating your forest land.

29. What levels of help do you think you need to increase your willingness to run your business?

A. Social dimension B. Economic dimension

C. Policy dimension D. Other dimensions

30. What suggestions do you have for the future development of forestry in ethnic areas?

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
