# Peer review of "Willingness and Influencing Factors of Farmers’ Forestland Management in Ethnic Minority Areas: Evidence from Southwest China"

_forests, doi:10.3390/f14071377_

Round 1
Reviewer 1 Report
This manuscript uses the Sustainable Livelihood Approach (SLA) framework to assess minority Forest Farmers' willingness to engage in forest management in two counties in Yunnan Province. Thank you authors for your hard work.
Before I can recommend this manuscript for publication, there are several major issues that must be addressed:
The Abstract is disconnected from the manuscript.
- In the Abstract (line 17), the authors raise the question "are there differences in the willingness and behavior of forest management between ethnic minority foresters and ordinary foresters?" This seems to be an interesting, and important, question, but is never addressed in the manuscript.
- In the Abstract (line 30) the authors report that literacy has a positive effect, yet in the Discussion they report a negative effect. This seems to be a major difference.
- In the Abstract (line 33) the authors report significant differences between minority and non-minority regions - with no evidence in the manuscript
- Many of the recommendations made by the authors (beginning line 35) do no seem to be supported by the model.
- Line 47 seems to be an unusual way to open the Introduction, with minimal direct relevance to the rest of the work.
- In the Introduction, an English language rewrite will help greatly. Line 81 (beginning Goyke et al.) is a especially egregious example of writing that left me unsure of what the authors were trying to say.
- In line 104, it may be helpful to immediately define the mu as a unit for international readers.
- Line 120, have the authors taken any considerations to address the potential gap between willingness and actual behavior?
- Line 149, the sentence beginning "It is found..." is especially difficult to understand.
- Line 156, how does this situation represent a "tragedy of the commons"?
Towards the end of the manuscript (line 543) the authors seem to imply that the data were collecting by other scholars, not the authors themselves, and they are doing an analysis of those data. It would be helpful to the reader to know more about that survey instrument, more about the authors' roles in data collection, and why use data that the authors admit is not designed to explore willingness to manage but instead on reform effectiveness.
- What is the % forest cover in Jianchuan County?
- The Legends in Fig. 1 and Fig. 2 are unclear
- What is the unit of the Legend in Fig. 3?
- Were the survey respondents representative of the greater population in terms of cadre identity, education, standard of living etc
- 98.38% is a very high survey response rate. How did the authors accomplish this?
- The SLA seems to be an appropriate framework for this analysis, as forest management can be a pathway to poverty reduction. However, it is not totally clear how the SLA (which emphasizes five types of capital) is related to the variables in the logistic regression model.
- Is there literature to support the three points beginning with line 282?
- The SLA framework (Fig. 4) includes operational conduct and efficiency, but neither a dependent variable in any model? Why not either run additional models, or else leave those out of the figure?
- How several of the variables in the model are defined is unclear. For example, cadre is undefined. How education and living standard were coded is unclear (for example, why would a person be coded 1 or 0 for X2?)
- X4 and X11 are reported at a different order of magnitude than the other 9 variables. Do they need to be transformed or normalized for inclusion in the logistic regression model?
- Lines 353 and 358 both begin Second(ly). Should 358 begin Third?
- In Table 4. it could help readers if the independent variables (argument) were defined in words: identity, education, etc. instead of as X1, X2, etc. It reduces the need to flip back and forth between pages.
- It is currently unclear what the relevance of the data presented in Tables 5 & 6 is to the authors' question of willingness to participate in forest management
- Several of the conclusions (for example lines the role of age in line 573, negative attitudes towards logging in line 602) are not supported by the research presented in the manuscript.
Overall, more clarity about the data, how they were collected, and the role of the authors in data collection are necessary. Additionally, the results, discussion, and conclusion must be more closely tied to the model, and avoid speculation.
The current version of the manuscript needs improvement in the Quality of English Language writing. In places the low quality English is distracting, and detracts from the reader's ability to understand what the authors are trying to say. In this current manuscript, I have given the quality of the authors' science and mastery of the literature the benefit of the doubt, but the writing must be improved before I can recommend this manuscript for publication.
Author Response
We are very grateful to your comments for the manuscript. Your comments and suggestions will play an important role in improving the quality of the manuscript. I have revised and improved this paper in strict accordance with your comments. All of your questions were answered one by one.( Please see the attachment.)
Point 1: In the Abstract (line 17), the authors raise the question "are there differences in the willingness and behavior of forest management between ethnic minority foresters and ordinary foresters?" This seems to be an interesting, and important, question, but is never addressed in the manuscript.
Response 1: Thank you for highlighting this deficiency. Of the 3 points of reflection on farmers in ethnic minority areas based on the interviews after the SLA framework was proposed in section 2.3 of the paper, the first of which explains the differences in willingness and behavior of ethnic minority foresters at the subjective level compared to ordinary foresters based on the farmers' perspective.
In addition, we added a paragraph to section 4.2 explaining the influence of farmers' perceptions and related experiences on their willingness to operate, and explained the contrasts and differences in behavior, philosophy, and cultural practices between ethnic minority foresters and ordinary foresters.
Point 2: In the Abstract (line 30) the authors report that literacy has a positive effect, yet in the Discussion they report a negative effect. This seems to be a major difference.
Response 2: Thank you for your reminder. We are very sorry for our careless mistake. During the discussion, it was concluded that "living standard" and "forest area" have a positive effect on willingness to operate, while "education level" has a negative effect on willingness to operate. When analyzing the results against the model in Table 4, the variables were not labeled with text, thus making my writing confusingly wrong and causing a bias in your understanding of the reported results, which I have corrected in the abstract according to the correct results.
Point 3: In the Abstract (line 33) the authors report significant differences between minority and non-minority regions - with no evidence in the manuscript.
Response 3: Thank you for your professional comments. In part 2.3 on the consideration of the second and third points for foresters in ethnic areas, the consideration of objective factors based on the geographical location and resource endowment of Yunnan Province indirectly affects the business behavior and willingness of farmers in ethnic areas. In addition, based on the interviews, it is summarized in 4.2 that the high degree of fragmentation of forest land in ethnic areas of Yunnan makes it more costly and difficult to operate in a concentrated, continuous and large-scale manner compared to non-ethnic areas, making farmers in ethnic areas less willing to operate. It is explained in 4.3 that the high proportion of public interest forests in ethnic areas and thus the policy restrictions prevent foresters in ethnic areas from cutting them down and the compensation standard is lower, which makes foresters' management behavior change greatly.
These situations and limitations rarely exist in non-ethnic areas, where large-scale and joint-family operations are more common, and where the proportion of public-good forests is very low, compared to the benefits derived from forest land management.
Point 4: Many of the recommendations made by the authors (beginning line 35) do no seem to be supported by the model.
Response 4: Thank you for your comment, our response is as follows:
- The results of the model are the conclusions drawn from a micro-level survey of farmers, and the conclusions drawn from the model can reflect the willingness and behavior of farmers in ethnic areas to manage forest land. However, in order to be clearer about the willingness and behavior of farmers to operate forest land in these areas, we also had interviews with large forest landowners, new management entities, local government and other relevant people during the survey (Appendix A). And these interviews are macro-level textual content, which cannot be processed in the model for variables and micro-analysis, but the business willingness they reflect also serves as an important reference part for the future development of forestry in ethnic areas, so the countermeasures we propose are the result of a combination of model analysis and interviews.
- With regard to the recommendation of "concentrated and continuous management and encouragement of capital investment", the variable "area of forest land" was also applied in the model, taking into account the content of the interviews and macro-level factors such as the geographic location of the survey site and the forest land resources. It was found that the forest land per capita in ethnic minority areas is large, but the management behavior and willingness are weak, probably due to the fine plot and low mechanization. Farmers do not have a large amount of capital to invest, so it is proposed whether we can encourage outside capital investment to help farmers achieve appropriate scale operation to improve returns in order to enhance the willingness to operate.
- The suggestion of "optimizing the harvesting limit system and raising the standard of public welfare forest" is related to the variable of "whether they are compensated by public welfare forest" in the model, and the reasons for not wanting to operate forest land, such as "low compensation standard" and "no harvesting target" in Table 6. Therefore, some of the recommendations made, although not directly correlated with several variables that have an impact in the model results, are integrated by considering the survey content, interview content, and micro-macro factors.
Point 5: Line 47 seems to be an unusual way to open the Introduction, with minimal direct relevance to the rest of the work.
Response 5: Thank you for pointing out this flaw. Due to a translation issue, there was an error in the content of the expression. We have done our best to touch up the language in the revised draft.
Point 6: In the Introduction, an English language rewrite will help greatly. Line 81 (beginning Goyke et al.) is a especially egregious example of writing that left me unsure of what the authors were trying to say.
Response 6: Thank you for pointing out this flaw. Due to a translation issue, there was an error in the content of the expression. We have finished the paper through the English editing service provided by MDPI. The polishing proof can be seen from the attachment.
Point 7: In line 104, it may be helpful to immediately define the mu as a unit for international readers.
Response 7: Thank you for your reminder and suggestion. We have converted the numbers in "mu" to "hectare" in the text, where 1 hectare = 15 mu.
Point 8: Line 120, have the authors taken any considerations to address the potential gap between willingness and actual behavior?
Response 8: Thank you for your question. This paper is intended to address several questions: Whether farmers in ethnic areas have strong desire to manage forest land?What are the factors influencing farmers' willingness to operate forest land in ethnic areas? Are there differences in the characteristics of woodland resources in ethnic areas compared to non-ethnic areas?” And the differences between farmers' willingness and behavior in forest land management are the focus and direction of our future research. Thank you again for providing us with good ideas and references for in-depth research in this area.
Point 9: Line 149, the sentence beginning "It is found..." is especially difficult to understand.
Response 9: Thank you for pointing out this flaw. Due to a translation issue, there was an error in the content of the expression. We have finished the paper through the English editing service provided by MDPI.
Point 10: Line 156, how does this situation represent a "tragedy of the commons"?
Response 10: Thank you for your question. Our answer is as follows: The "tragedy of the commons" is mainly used as a metaphor for the situation in which limited resources are used freely without restriction and eventually lead to excessive exploitation of resources, in which one of the main reasons is human selfishness and another reason is the inequality in the process of conversion of individual and public interests. In this paper, we use the example of the "tragedy of the commons" model to illustrate that if forest resources are left unused, they may also become wasteland, stranded, and degraded. Ultimately, the results are similar to those of the "tragedy of the commons" example, in that they do not allow the resources to be used rationally and are not conducive to the sustainable use of forest resources, thus raising the question of how to fully and effectively use forest resources.
Point 11: Towards the end of the manuscript (line 543) the authors seem to imply that the data were collecting by other scholars, not the authors themselves, and they are doing an analysis of those data. It would be helpful to the reader to know more about that survey instrument, more about the authors' roles in data collection, and why use data that the authors admit is not designed to explore willingness to manage but instead on reform effectiveness.
Response 11: Thank you for pointing out this flaw. Since we made a mistake in our presentation and translation, we offer the following explanation: The research topic is conducted in the context of the reform of collective forest rights system in Yunnan Province, to which we have added in the introduction section. In order to study the utilization of forest land resources after the reform, the evaluation team of the Peking University research project commissioned each province to conduct a survey of relevant questions. The questionnaires were standardized and normalized before being used. At the same time, our team was the trustee, in which Ms. Li Ya was the project leader, and both Haiqing Chang and Yaquan Dou were involved in the questionnaire and data collection.
In addition, reform is a continuous push and not an ephemeral project. The forestry industry is also a dynamic development industry that requires constant identification of problems so that the government, the Forestry Bureau, the state and other relevant departments can develop policies to solve the problems. It is also because of our participation in this research activity that we found this interesting research question.
Point 12: What is the % forest cover in Jianchuan County?
Response 12: Thank you for your question. Due to our oversight, we did not state the forest coverage rate of Jianchuan County in the article, we have now added in 2.1 that the forest coverage rate of Jianchuan County is 74.5%.
Point 13: The Legends in Fig. 1 and Fig. 2 are unclear
Response 13: Thank you for pointing out this defect. We have reworked Figures 1 and 2 to make the legends more clearly visible.
Point 14: What is the unit of the Legend in Fig. 3?
Response 14: Thank you for your question. In Figure 3, the units in the legend section are: billion dollars. The darker color in the graph is based on the forestry output value of each prefecture-level city or prefecture in Yunnan Province. The darker color represents the higher forestry output value of the region, which can represent the output of all forestry products in monetary terms in the region in a certain period of time, and also represents the level of development of forestry industry and resource utilization in the region. Also, we have redone Figure 3 so that the reader can see the legend more clearly.
Point 15: Were the survey respondents representative of the greater population in terms of cadre identity, education, standard of living etc
Response 15: Thank you for your question. Due to limited research time and funding, the group of farmers in ethnic areas is very large, and the use of a comprehensive survey is not realistic, and the work cannot be completed in a short period of time. Therefore, this survey adopts the form of sampling (most of the field survey projects are completed in the form of sampling), selected in accordance with the principle of random, selected groups for investigation, and based on this to make estimates and inferences for all respondents, the purpose is to obtain information reflecting the overall situation, and therefore play a comprehensive survey. Regarding the variables such as cadre status and education level, which are the contents of the survey respondents' answers to the questionnaire about personal information, we organized them into relevant variables for model evaluation, and the variables can reflect the situation of the survey respondents, and the survey respondents can represent the overall level. At the same time, we also made additional explanations in section 2.2.
- Each minority farmer has an equal chance of being sampled, thus ensuring that the probability of survey respondents being sampled is equal and evenly distributed in the overall population, with no bias error and a high degree of representativeness.
- The sample size of the selected minority foresters is also determined by scientific calculation according to the requirement of survey error, and the number of samples in the survey is guaranteed to be reliable.
- The error of the survey sample is calculated before the survey based on the sample size and the degree of variation between the overall sample and controlled within the allowable range, and the accuracy of the survey results is higher.
Point 16: 98.38% is a very high survey response rate. How did the authors accomplish this?
Response 16: Thank you for your professional comments on our article. Our answer is as follows: First of all, before the survey was conducted, we conducted professional interviews and training for the team members in advance in order to improve the efficiency of the questionnaire collection, and to help the team members to understand the specific meaning and significance of each question in the questionnaire. All team members have participated in social survey projects and have sufficient experience. Also, team members' majors are related to agriculture and forestry, which helps members to guide farmers to answer questions correctly to a large extent (briefly explained in section 2.2). Secondly, in the actual conduct of the survey work, there are village leaders (in the hearts of farmers more prestigious personnel), the assistance of the relevant leaders of the Forestry Bureau to guide the surveyed foresters to actively cooperate with this survey. Finally, after the completion of our team's survey work, we organized the data of the questionnaires and made a return visit to the relevant farmers regarding the questionable data to improve the recovery rate and the quality of the questionnaires. As a result, we had a high response rate and effective rate, and only 3 out of 185 questionnaires were invalid.
Point 17: The SLA seems to be an appropriate framework for this analysis, as forest management can be a pathway to poverty reduction. However, it is not totally clear how the SLA (which emphasizes five types of capital) is related to the variables in the logistic regression model.
Response 17: Thank you for your professional comments on our article. Due to our oversight, the relationship between the variables in the model and several types of capital in Figure 4 was not explained clearly when introducing the SLA framework. We have refined and added in Figure 5 about the correlation between the SLA framework and the variables so that the reading public can more clearly understand the association between the SLA framework and the variables in the model.
Point 18: Is there literature to support the three points beginning with line 282?
Response 18: Thank you for your comments. We have summarized it based on reading the book 《Supplementary Notes on Yunnan Zhi》. In addition, interviews were conducted with ethnic minority foresters, experienced elderly villagers, village cadres, and other personnel, and three points of reflection on foresters in ethnic areas were summarized, leading to research hypotheses that are helpful for variable selection and model construction. Regarding the references, we have added and marked them in the new revised manuscript.
Point 19: The SLA framework (Fig. 4) includes operational conduct and efficiency, but neither a dependent variable in any model? Why not either run additional models, or else leave those out of the figure?
Response 19: Thank you for your professional comments on our article. Due to our oversight, the relationship between the variables in the model and the several types of capital in Figure 4 was not explained clearly when introducing the SLA framework. We have refined and added the correlation between the SLA framework and the variables in Figure 5. as shown in Figure 5, so that the reader can more clearly understand the connection between the SLA framework and the content of our study. The purpose of the study and the research hypothesis need to be presented before the study. The SLA framework provides us with the idea of dividing the variables, and we can only analyze and run the model with the actual, assigned variables from the questionnaire in the SPSS software.
In addition, business behavior and efficiency in Figure 4 is not the main purpose of this paper's research, in the will to translate into behavior and efficiency still need some analysis, thank you for your professional comments and guidance. After our discussion, we decided to adjust the content of Figure 4 by deleting the keywords "business behavior" and "business efficiency" and keeping "business intentions" as the main object of study. In addition, other expressions related to "business behavior" were also adjusted and revised.
Point 20: How several of the variables in the model are defined is unclear. For example, cadre is undefined. How education and living standard were coded is unclear (for example, why would a person be coded 1 or 0 for X2?)
Response 20: We sincerely thank the reviewers for their careful reading. There are three data types: Nominal, Ordinal, and Scale, when SPSS software processes data for model fitting. They correspond to: fixed-class data, fixed-order data, and fixed-distance data, respectively. For example, cadres, education, and living standards are classifications of different individuals of foresters in ethnic areas according to different characteristics, which belong to definite data. For example, the answer to the question "status of farmers" includes two kinds of answers, including village cadres and ordinary people, which is a text type answer and has no numerical value in itself. The SPSS software does not recognize textual variables and cannot perform calculations, so it is necessary to assign values to "village officials" and "ordinary people" and convert them into numerical forms that can participate in operations.
Point 21: X4 and X11 are reported at a different order of magnitude than the other 9 variables. Do they need to be transformed or normalized for inclusion in the logistic regression model?
Response 21: We sincerely thank the reviewers for their careful reading. x4 and x11 represent the area of forest land and the rating of the harvesting limit policy, respectively. both variables are quantitative variables (numerical variables), which are variables that describe numerical information about things, and the variable values themselves are numbers. Unlike the textual answers such as "farmer status" and "education level", there is no need to specialize or assign values to these answers, and they can be directly referenced to the logistic model. The fundamental difference between numeric variables and the above-mentioned subtype variables is that numeric variables are meaningful for addition, subtraction, averaging, etc. or in the calculation of models; whereas subtype variables are not meaningful for some numbers or model calculations that cannot be identified.
Point 22: Lines 353 and 358 both begin Second(ly). Should 358 begin Third?
Response 22: Thank you for highlighting this defect. We are very sorry for our careless error. In the original manuscript, the two paragraphs represented by lines 353 and 358 were identical due to our translation and writing errors, and we have now made the changes. Thank you again for the heads up.
Point 23: In Table 4. it could help readers if the independent variables (argument) were defined in words: identity, education, etc. instead of as X1, X2, etc. It reduces the need to flip back and forth between pages.
Response 23: Thank you for highlighting this deficiency. We have made a correction in Table 4 to define the variables represented by letters such as X1 and X2 in words so that it is easier for us to fix the content and for other readers to read in. Thank you again for your tips and comments.
Point 24: It is currently unclear what the relevance of the data presented in Tables 5 & 6 is to the authors' question of willingness to participate in forest management.
Response 24: We appreciate your professional comments on our article. In order to explain more clearly the relationship between Table 5 and foresters' willingness to operate, we have added and expanded the content in both part 2.2 and part 3.2 of the article. The reform of collective forest rights system in Yunnan Province can activate the forest farmers' land management rights and place the power of disposal of resources at the macro level, while the core of the forest rights (forest power) system is forest land, and its reform can make farmers become the main body of forest rights management, which will constrain the farmers' management behaviors such as transferring, contracting and leasing of forest land. Forest land titling is not only conducive to solving the problem of asymmetry between forest utilization rights and protection responsibilities, but also realizes the goal of forests serving farmers' income increase and farmers contributing to sustainable forest management. It affects farmers' enthusiasm to get rich and plant trees, thus directly influencing farmers' management behavior and their willingness to operate in the future.
Table 6 shows the descriptive statistics on farmers' willingness to operate. The explanation shows that 28.2% of the 182 foresters do not have the willingness to operate the forest land, and the reasons such as "low income, low compensation standard of public welfare forest, high operating cost due to fragmentation of land" are given. These textual reasons intuitively indicate farmers' business attitudes, and the text cannot be translated into variables in the model, but directly affects the central content of our study. The fact that 71.98% of the foresters have the willingness to run a business, which is a high percentage, has stimulated us to think about the willingness of farmers in ethnic areas, and to explain the differences with non-ethnic areas and ordinary foresters in the "Discussion" section later.
Point 25: Several of the conclusions (for example lines the role of age in line 573, negative attitudes towards logging in line 602) are not supported by the research presented in the manuscript.
Response 25: We sincerely appreciate the valuable feedback from the editors and all reviewers, which we use to improve the quality of the manuscript. As you were concerned, there were several issues that needed to be addressed. Based on your suggestions, we have made extensive corrections to the "Conclusions" section of the previous manuscript.

Reviewer 2 Report
Manuscript entitled "Are Minority Forest farmers Willing to Participate in Forest Management ? Evidence from the Minority Regions of South-
west China" is interesting and raises important issues of forest land management in China.
1. It is worth modifying/refining the title of the study to make it more interesting and unambiguous.
2. It is worth specifying the conclusions in the context of the research goals set.
Author Response
Thank you very much for your affirmation and consideration of this manuscript. Your comments and suggestions on this manuscript will play a great role in improving the quality of the manuscript. We have revised it according to your comments and suggestions. Best wishes for you.
Point 1: It is worth modifying/refining the title of the study to make it more interesting and unambiguous.
Response 1: Thank you for your suggestions. We have done our best to touch up the title and language in the revised manuscript, and we sincerely thank the editors and reviewers for their enthusiastic work and hope that the corrections will be approved.
Point 2: It is worth specifying the conclusions in the context of the research goals set.
Response 2: We appreciate your professional comments on our article. Thank you for highlighting this flaw, which we think is a good suggestion. We have followed your suggestion and have presented the objectives and background of the study in the abstract section and added background and purpose about the reform, investigation in the introduction section to make the conclusions of the article more convincing.

Reviewer 3 Report
The authors are analyzing very important topic. By actively participating in forest management, landowners can contribute to the well-being of their forest, conserve biodiversity, and promote sustainable land use practices.
However, there are several areas that require attention and revision before it can be considered for publication.
-
Clarify Research Objectives: The paper lacks a clear statement of research objectives. It is essential to explicitly state the purpose of the study and any specific research questions or hypotheses to guide the reader throughout the paper.
-
Methodology: The paper lacks a description of the methodology employed to gather data on the willingness of forest farmers to participate in forest management. The authors need to specify the research methods, sample size, and data collection techniques employed. This information is crucial for assessing the validity and reliability of the findings. Additionally, a clear explanation of how forest owners were identified and recruited for the study would enhance the transparency and replicability of the research.
- Conclusions: The paper would benefit from a section dedicated to providing recommendations for forest farmers interested in participating in forest management.

Minor language editing is needed
Author Response
Thank you for your careful review. We appreciate your efforts in reviewing our manuscript during this unprecedented and challenging time. We wish you, your family and community good health. Your careful review has helped to make our research clearer and more comprehensive, and our point-by-point responses are listed above. Responses to your comments are highlighted in red in newly submitted revisions, and responses to questions are provided one-by-one below.
Point 1: Clarify Research Objectives: The paper lacks a clear statement of research objectives. It is essential to explicitly state the purpose of the study and any specific research questions or hypotheses to guide the reader throughout the paper.
Response 1: We appreciate your professional comments on our article. Thank you for highlighting this flaw, which we think is a good suggestion. We have followed your suggestion and have presented the objectives of the study in the abstract section and added information about the reform, the background and purpose of the investigation in the introduction section, making the conclusions of the article more convincing.
Point 2: Methodology: The paper lacks a description of the methodology employed to gather data on the willingness of forest farmers to participate in forest management. The authors need to specify the research methods, sample size, and data collection techniques employed. This information is crucial for assessing the validity and reliability of the findings. Additionally, a clear explanation of how forest owners were identified and recruited for the study would enhance the transparency and replicability of the research.
Response 2: Thank you for pointing out this deficiency. In order to improve the transparency and comprehensibility of the study, we have reintroduced additions and corrections to the survey methodology, sample size, and data collection techniques in section 2.2 to guide the reading public to more easily understand the survey process.
Point 3: Conclusions: The paper would benefit from a section dedicated to providing recommendations for forest farmers interested in participating in forest management.
Response 3: We appreciate your professional comments on our article. Thank you for highlighting this shortcoming, which we think is a good suggestion. At present, in addition to giving suggestions to address the conversion of foresters' business intentions and business behaviors in ethnic areas from the government's perspective, we have added additional suggestions about the foresters' perspective in the conclusion section, taking into account the results of the model analysis and the interviews.
Respond to Specific comments
Point 1 : 185 or 182 as it is stated in Methods section?
Response 1: We are very grateful for your professional comments on our article. We surveyed 185 minority farmers in the actual survey, so a total of 185 electronic questionnaires were collected, and we returned to the foresters again by collating the information to improve the accuracy and validity of the questionnaire data. The data from three questionnaires could not be used in the model in the end, the foresters answered logically contradictory questions and could not be contacted during the phone call back, so the data from these three questionnaires were excluded as invalid questionnaires, and the base in the logistic model was shown as 182.
Point 2 : The paper is missing clearly defined aim and/or research questions. Authors must add this in the introduction.
Response 2: We sincerely thank you for your valuable comments. We think it is a good suggestion. We have added relevant content about the purpose of the study and the main questions of the study in the introduction and abstract sections.
Point 3 : “get rid of poverty and become rich”: this statement is too strong and one reference is absolutely not enough to support it. also, the reference does not refer to the income of private forest owners. either support this statement with relevant research or delete it. the poverty of rural areas is a particularly sensitive topic and it is irresponsible to claim that just by owning a forest there is a possibility for landowners to get rich.
Response 3: Thank you for pointing out this flaw. We apologize for our carelessness, which resulted in an error in the expression due to a translation issue. We have done our best to touch up the language in the revised draft to correct this absolute statement.
Point 4 : In line 100 of the original manuscript, add reference that support this statement / data
Response 4: We are very grateful for your valuable comments. We have carefully reviewed the literature and have added references to support this section in the revised manuscript.
Point 5: what is this? --“the management status of forest landowners”
Response 5: Thank you for your question. Due to a problem with our translation, the expression was incorrect. We have removed this misrepresentation. We have also done our best to touch up the content of the article that did not make sense, so thank you very much for pointing out this flaw.
Point 6 : what do you mean by this? what is "modern forestry"?
Response 6: We appreciate your professional comments on our articles. Forestry modernization is a state or process in which the factors and combinations of forestry production undergo continuous change, evolving, constantly updating and developing. Its basic features are diversification of forestry functions, scientific operation, information management, mechanization of equipment and quality service.
Point 7 : something is wrong with these sentences. is this 1 or 2 sentences* please check the English here “evaluation project team of Peking University on August 212 23-September 4, 2021 to conduct a field The survey was conducted mainly”.
Response 7: Thank you for pointing out this flaw. Due to a translation issue, there was an error in the content of the expression. We have finished the paper through the English editing service provided by MDPI. The polishing proof can be seen from the attachment.
Point 8 : what was the purpose of these interviews? are collected data shown in section 3? please add detailed description of how the respondents were selected.
Response 8: We are very grateful for your professional comments on our article. First, regarding the purpose of the interview, we have rewritten and added to the abstract and introduction sections of the revised manuscript. Second, the collected data are presented in Table 1, Table 2, and Section 3.2 of Part III, and we standardized the collected data and assigned variables for better operation in the model. Finally, we added the method of selecting respondents, and because of the large overall sample size, a random sampling method was used to select foresters in ethnic areas. (1) Each minority farmer has an equal chance of being sampled, thus ensuring that the probability of survey respondents being sampled is equal and evenly distributed in the overall population, with no bias error and a high degree of representativeness. (2) The sample size of the selected minority foresters is also determined by scientific calculation according to the requirements of survey error, and the number of samples in the survey is guaranteed to be reliable. (3) The error of the survey sample is calculated before the survey based on the sample size and the degree of variation between the overall sample and controlled within the allowable range, and the accuracy of the survey results is higher.
Point 9 : interview” or survey? please, do not use interview and survey as synonyms.
Response 9: We are very grateful to you for raising this defect. We apologize for our carelessness, which resulted in incorrect expressions due to translation problems. The research is both based on the part of the questionnaire, which is mainly used for model analysis. There are also sections of interviews with elderly villagers, forestry leaders, village and town leaders, and other people, which are shown in the three reflections on farmers in ethnic areas and in the conclusion section. We have made corrections based on your suggestions, and we have done our best to touch up the language in the revised version.
Point 10 : How it was decided that 185 are enough? what were the criteria for selecting the final number of surveys?
Response 10: We would like to thank you for your professional comments on our article. We conducted a residual test on 182 valid questionnaires and found a normal distribution (as shown in the figure), indicating that the residuals meet certain theoretical requirements, so the sample size is sufficiently representative and the model built on 182 samples is reasonable enough.
It is generally considered that more than 50 samples can be considered as a large sample size with a certain overall representativeness, which is capable of doing logistic models. In addition, before the questionnaire work was carried out, we conducted professional interviews and training for the team members in advance in order to improve the efficiency of the questionnaire recovery, to help the team members to understand the specific connotation and meaning of each question of the questionnaire, the team members have participated in social survey projects and have sufficient experience, at the same time, the team members' majors are related to agriculture and forestry, which helps the members to guide correctly to a large extent We can ensure the quality and efficiency of the questionnaire.
Due to the limited research time and funding, the group of farmers in ethnic areas is very large, and it is not practical to use the form of a comprehensive survey, and the work cannot be completed in a short period of time. Therefore, this survey adopts the form of sampling survey (most of the field survey projects are completed in the form of sampling survey), selected according to the principle of randomness, some groups are selected for investigation, and based on this to make estimates and inferences for all the respondents, the purpose of which is to obtain information and materials reflecting the overall situation, and thus play the role of a comprehensive survey. Finally, in addition to the residual test, our research subjects were farmers, and we encountered many problems during the research period: farmers working outside the home, difficulty in understanding minority languages, etc. It was our team's best effort to collect 182 samples.
Point 11 : more appropriate title is:“Data analysis” This whole section 2. is dedicated to research methodology.
Response 11: We are very grateful for your professional comments on our article. We think this is a good suggestion and have therefore changed the original title of 2.4: Research Methodology to the more appropriate title: Data analysis.
Point 12 : add references to support these statements.
Response 12: We appreciate your professional comments on our article. We sincerely appreciate your valuable comments and, as you suggested, we have combined the results of the model analysis with the interviews and added references to support these in "4. Discussion".

Round 2
Reviewer 3 Report
Authors accepted all reviewers' comments. This demonstrates a commitment to enhancing the quality and credibility of their research. It also reflects a collaborative and constructive approach in the publishing process.
Author Response
Thank you for your careful comments. We really appreciate your efforts in reviewing our manuscript during this unprecedented and challenging time. We wish good health to you, your family, and community. Your careful review has helped to make our study clearer and more comprehensive, and our point-by-point responses are presented following. The responses to your comments are marked in red and presented following.
Point 1: The concept of "forest quality" is not clear in the manuscript, especially in the sentences where the objectives have been shown ("the precise improvement in forest quality in ethnic minority areas")
Response 1: Thank you for your professional comments. In the abstract, we wrote "precise improvement of forest quality in ethnic minority areas", and the improvement of forest quality includes achieving a virtuous industry-ecosystem cycle in forestry in ethnic areas, preserving and increasing the value of natural forest assets, and stimulating the endogenous motivation of ethnic minority foresters to participate in forestry industry and ecological construction, etc. Since we do not express ourselves clearly, the purpose of the study has now been presented in a more refined and specific manner, and changes have been made in the newly revised manuscript.
Point 2: The questionnaire (in the English language) should be incorporated as an Annex or Supplementary Material.
Response 2: Thank you for highlighting this flaw. In order to give readers a clearer understanding of the article, we have extracted questions related to the research content of the article and we have added a questionnaire in the form of an appendix in the newly revised version. Thank you again for your professional opinion.
Point 3: In my view, the explanation of Figure 5 is based on grey literature. One of the most cited papers regarding environmental valuation (http://dx.doi.org/10.1016/j.gloenvcha.2014.04.002) shows another way to consider the different forms of capital.
Response 3:Thank you for your professional comments. Comparing your recommended literature and our article, we found that the purpose and content of the two studies are different. (1) The reference literature is based on a macro-level finding that ecosystem services refer to the relative contribution of natural capital to human well-being, that natural capital does not flow directly into human well-being capital, and that we need to approach ecosystem services with a broad, interdisciplinary perspective. Examining the value of ecosystem services by considering different forms of capital leads us to think about how to create a more comprehensive and balanced interrelationship between assets and people. (2) Our article is based on the micro level and divides the variables in the model by the five categories of capital included in the SLA framework, so as to investigate the micro level influences that affect foresters' willingness to operate. Therefore, in combination with experts' recommendations, and to give the reader a clearer picture of which category of capital the variables belong to, we indicate in Figure 5.
In the meantime, we thank you again for your comments. In our future more in-depth studies, we will carefully read and consider the novel forms of this article on the application of different capitals to ecosystem studies. We will also consider a more macro perspective on ecological issues to make the study more precise and meaningful.
Point 4: The Conclusions Section should be shortened, avoiding sentences not directly related to this manuscript's objectives and results.
Response 4:Thank you for your professional opinion. We think your suggestions will be helpful to improve the quality of the manuscript. In the previous manuscript, we also stated some recommendations for promoting forestry in ethnic areas in the conclusion section. We have streamlined the conclusion section by correcting and appropriately deleting sentences that are not directly related to the objectives and results of this manuscript.
